# Effect of Drying Methods on Volatile Compounds of *Citrus* *reticulata* Ponkan and Chachi Peels as Characterized by GC-MS and GC-IMS

**DOI:** 10.3390/foods11172662

**Published:** 2022-09-01

**Authors:** Xiangying Yu, Xiaochun Chen, Yuting Li, Lin Li

**Affiliations:** Engineering Research Center of Health Food Design & Nutrition Regulation, School of Life and Health Technology, Dongguan University of Technology, Dongguan 523808, China

**Keywords:** citrus peel, volatile compounds, sun-drying (SD), hot-air-drying (AD), freeze-drying (FD), GC-MS, GC-IMS

## Abstract

To reflect the volatile differences of dried citrus peel as affected by cultivars and drying methods, the volatile compounds of dried citrus peel of two cultivars (*Citrus reticulata* “Chachi” and *Citrus reticulata* “Ponkan”), prepared under three drying methods (sun-drying (SD), hot-air-drying (AD), and freeze-drying (FD)), were analyzed by GC-MS, odor activity values (OAVs), and GC-IMS. GC-MS data indicated that SD was favorable to preserve terpenic alcohols (linalool, α-terpineol and terpinene-4-ol), β-cymene, methyl methanthranilate, and monoterpenes; while AD was favorable to preserve aliphatic aldehydes and sesquiterpenes; and SD was more similar with AD in GC-MS analysis of volatile profile (of higher MW) for both cultivars from the PCA outcome. Furthermore, significant difference in volatile isomeric composition of different samples was also clearly demonstrated through extracted ion chromatogram (EIC) by GC-MS analysis. GC-IMS analysis showed the favorability of FD to preserve ketones, phenols, esters, and aromatic aldehydes; and SD was more similar with FD in GC-IMS analysis of volatile profile (of smaller MW) for both cultivars from the PCA outcome. Moreover, the OAVs indicate that 2-methoxy-4-vinylphenol contributed much to the flavor of dried Ponkan peel, while 2-methoxy-4-vinylphenol, methyl methanthranilate, and methyl anthranilate played an important role in the flavor of dried Chachi peel; and the highest OAVs for monoterpenes were observed at SD for both cultivars. Thus, the combination of GC-MS and GC-IMS analyses with PCA in this paper suggested the superiority of SD to preserve volatiles and characteristic aroma in dried citrus peel, and that SD contributed much to the quality of dried Chachi peel.

## 1. Introduction

Citrus, belonging to the Rutaceae family, are well known as one of the world’s major fruit crops that are produced in many countries, such as China, USA, Brazil, etc. [1,2]. Due to the flavorful and pro-health properties, citrus fruits are one of the most readily consumed fruits in the world. In addition to large scale consumption as fresh fruits, the citrus fruits are mainly processed to produce juice, which generates a huge amount of citrus waste [3].

Citrus peel, the main byproduct of the citrus industry, has rich bioactive compounds, including flavonoids, phenolic acids, volatile oil, etc. [4], which were reported to possess health-promoting benefits including antioxidant, antifungal, antibacterial, hypolipidemic effects [5,6,7]. Thus, the fruit peels resulted from juice processing, were fully utilized in food industry, cosmetics, and folk medicine. With the risk of being prone to microbial spoilage, the citrus peel was often dried before being utilized as culinary seasoning, dietary supplement, and food additive [8] for its dietary values in smell, flavor, and curative effects [4], and as the herbal medicines for treating cough, indigestion, inflammatory respiratory diseases, and others [9]. Since the citrus plant and fruit were prone to pest infection, a multitude of pesticides were used for citrus fruits during the cultivation process [10]; thus, the danger of dried citrus peel from pesticide residues should not be ignored.

For dried citrus peels, most of the literature reports focused on flavonoid content [11]; however, aroma was also the important quality characteristic affecting consumer acceptance [12,13]. Phytochemical studies demonstrated that approximately 160 volatile compounds have been identified from dried citrus peel, including monoterpenes, sesquiterpenes, and *O*-containing compounds (acids, esters, aldehydes, ketones, and alcohols, etc.), with monoterpenes of the highest content among different cultivars [14,15]. In addition, for dried peel of citrus from different cultivars or origins, the volatile composition was often different. In fact, the volatile profile has been widely studied for classification of citrus species and citrus taxonomy [16,17,18].

Generally, the dried peel of *Citrus reticulata* “Chachi” from Xinhui County (Jiangmen, Guangdong, China) has always been well-regarded to have superior quality when being compared with dried peel derived from other cultivars with respect to geo-herbalism [19,20]. Thus, recently, under the economic adulteration that the dried Chachi peel was often superseded by cheaper or inferior variants in the market, the volatile profile of dried citrus peel has also been studied to develop accurate methods differentiating the dried peel of Chachi and other cultivars [14,21,22]. The preliminary market survey in China showed that the dried peel of Chachi was mainly prepared by natural sun-drying, while the dried peel of other citrus cultivars was mainly prepared by hot-air-drying due to the climate differences in different production areas. Therefore, for dried peel of different citrus cultivars, which were made from different drying methods, it is necessary to explore the effect of drying methods.

Principal component analysis (PCA), a multivariate statistics-based detection method, could be used to transform multiple indicators into a few comprehensive indicators for extracting features and revealing the relationship between variables. Recently, PCA has been widely used in food classification [23], such as identification of significant organic substances in raw materials or food products [24], as well as the demonstration of changes in food or raw material under the influence of technological processes [25,26].

Thus, in this study, fresh *Citrus reticulata* “Chachi” (Xinhui District, Jiangmen, Guangdong, China) and *Citrus reticulata* “Ponkan” (Jian ‘ou County, Nanping, Fujian, China) were collected to prepare dried peel by using three drying methods, that is, sun-drying, hot-air-drying, and freeze-drying. GC-MS and GC-IMS were used together to characterize the volatile compounds of dried citrus peel, and the odor activity values (OAVs) were used to characterize the aroma attributes. The volatile compounds characterized by GC-MS and GC-IMS representing different dried citrus peel samples were then subjected to PCA to analyze and generate a flavor fingerprint for dried citrus peel from different cultivars and drying methods. The aim of the present study was to decode the volatile differences of citrus peel as affected by two cultivars and three drying methods.

## 2. Materials and Methods

### 2.1. Plant Materials

The samples used in the experiment were two cultivars collected from two farms in November, 2020, that is, fresh *Citrus reticulata* “Chachi” from Xinhui Distrcit, Jiangmen City, Guangdong Province and *Citrus reticulata* “Ponkan” from Jian ‘ou County, Nanping City, Fujian Province. After cleaning, the citrus peel was removed as three pieces through three knife cuts and preserved at 4 °C before drying.

### 2.2. Reagents and Chemicals

A standard series of saturated alkanes C8–C40 and n-ketones C4–C9, used for retention index (RI) determination, was purchased from Supelco (Bellefonte, PA, USA) and Sinopharm Chemical Reagent Beijing Co., Ltd. (Beijing, China), respectively. All reference standards used for identification, including linalool, terpinen-4-ol, α-terpineol, citronellol, β-cymene, 2-isopropyl-5-methylanisole, thymol, carvacrol, methyl anthranilate, methyl methanthranilate, furfural, benzaldehyde, octanal, decanal, undecanal, dodecanal, (-)-α-pinene, (-)-β-pinene, β-myrcene, α-terpinene, D-limonene, γ-terpinene, terpinolene, α-farnesene, neryl acetate, methyl palmitate, methyl linoleate, were purchased from Aladdin reagent (Shanghai, China) Co., Ltd.

### 2.3. Drying of the Fresh Citrus Peels

About 100 g fresh citrus peel pieces were spread out evenly and subject to three different drying methods (sun-drying, hot-air-drying, and freeze-drying), and drying lasted until there was no change on the weight of peel. Finally, the water content (wt/wt) of the dried citrus peel prepared by different drying methods was all at the level of about 9%.

In sun-drying (SD), the fresh citrus peel pieces were left under direct sun daylight at temperatures about 25 °C for 4 days with about 32 h of daylight. In hot-air-drying (AD), the fresh citrus peel pieces were dried in an electric thermostatic drying oven (DHG-9240A; Shanghai Qixin Scientific Instrument Co., Shanghai, China) at 70 °C (the temperature was chosen based on the survey of production of dried Ponkan peel from Fujian province) for about 4 h. In freeze-drying (FD), the fresh citrus peel pieces were first frozen at −80 °C for 8 h, and then were quickly placed into a freeze dryer (Scientz-10N, Ningbo Xinzhi Biotechnology Co., Ningbo, China) and dried under vacuum (−60 °C, 1 Pa) for about 48 h. All dried citrus peel, that is, sun-dried peel from *Citrus reticulata* “Chachi” (Guangdong) (GSD) and *Citrus reticulata* “Ponkan” (Fujian) (FSD), hot-air-dried peel from *Citrus reticulata* “Chachi” (Guangdong) (GAD) and *Citrus reticulata* “Ponkan” (Fujian) (FAD), and freeze-dried peel from *Citrus reticulata* “Chachi” (Guangdong) (GFD), and *Citrus reticulata* “Ponkan” (Fujian) (FFD) were all stored at −20 °C until further analyzed.

### 2.4. Analysis of Volatile Compounds

#### 2.4.1. HS-SPME-GC-MS Analysis

HS-SPME-GC-MS analysis of dried citrus peel samples was conducted according to the method of Qiu et al. [27] with some modifications. A certain amount of samples (2 g) was finely ground in high-speed universal grinder (FW100, Tianjin Tester Instrument Co., Tianjin, China.) and mixed with 100 µL of phenyl ethyl acetate (20 µL/mL in methanol, an internal standard), then immediately sealed in a 20 mL vial covered with an aluminum seal with a PTFE septum. The sample vial was then equilibrated at 90 °C for 5 min on a heating platform. The extraction was conducted by inserting the preconditioned SPME fiber (DVB/PDMS) (75 μm) into the head space of the vial for 30 min at 90 °C. At the end of extraction, the fiber was desorbed into the injection port of GC for 5 min.

GC-MS analysis was performed on Agilent 8890/7000D (Agilent technologies Inc., Santa Clara, CA, USA). Volatiles were separated using a HP-5MS (30 m × 0.25 mm × 0.25 μm, Agilent technologies Inc., Santa Clara, CA, USA). The carrier gas was helium (>99.999%) at a constant flow rate of 1 mL/min. The oven temperature was initially held at 70 °C for 2 min, then increased to 210 °C at 4 °C/min, and held for 10 min. The injector temperature was maintained at 250 °C with a split ratio of 1:20. The transfer line to the mass spectrometer was maintained at 250 °C and the ion source temperature was 230 °C. The mass detector was operated in the electron impact mode with ionization energy of 70 eV and a scanning rage of 35–500 a.m.u. Compounds were identified by comparing their mass spectra and linear retention index (LRI) [28] with those contained in the NIST mass spectrum database, as well as the available reference standards. The content of the volatiles was approximately semi-quantified by comparing the peak area with that of the internal standard (phenyl ethyl acetate) in the total ion chromatogram. The peak area normalization method was also used to obtain the relative percentage content of each compound in the samples.

#### 2.4.2. HS-GC-IMS Analysis

HS-GC-IMS analysis was completed on a GC-IMS instrument (FlavourSpec^®®^, Gesellschaft für Analytische Sensorsysteme, Dortmund, Germany) equipped with an autosampler unit (CTC Analytics AG, Zwingen, Switzerland), which uses a 1 mL air-tight heated syringe to directly sample from the headspace. The GC was equipped with a FS-SE-54-CB-1 capillary column (15 m × 0.53 mm ID, 1 µm). Sample analysis was determined based on the method reported in the literature [29] with some slight modifications. A total of 0.5 g of finely ground samples were transferred into a 20 mL headspace glass sampling vial and incubated at 60 °C for 20 min with rotation speed of 500 r/min. A total of 500 µL of headspace samples was automatically injected into the injector (85 °C, splitless mode) by means of a heated syringe at 85 °C. Nitrogen (99.999%) was used as carrier gas and its flow rate was first set at 2 mL/min for 2 min, then increased to 10 mL/min within 8 min, then increased to 100 mL/min within 10 min, and then increased to 150 mL/min within 5 min, and held for 20 min. The analytes were eluted and separated at 60 °C, then driven into an ionization chamber (quasi-enclosed radioactive material) and ionized by a ^3^H ionization source in positive ion mode. The resulting ions were driven to a drift tube (98 mm in length) which operated on a constant temperature (45 °C) and voltage (5 kV). The retention index (RI) of each compound was calculated using n-ketones C4–C9 as external references. Volatile compounds were identified based on RI and drift time compared to GC-IMS library (NIST and IMS databases).

#### 2.4.3. Calculation of Odor Activity Values (OAVs)

To assess the attribution of the volatile compounds to the overall dried citrus peel aroma, the odor activity values (OAVs) were calculated by dividing the concentrations of aroma compounds with their sensory thresholds from the literature [30]. Only the compounds with an OAV greater than 1 contribute individually to the dried citrus peel aroma.

### 2.5. Statistical Analysis

The results given in the tables and figures were means and standard deviations of triplicate experiments. Experimental data were analyzed using Origin 8.0 (Microcal Software, Inc., Northampton, MA, USA). Significant differences between samples were analyzed using analysis of variance (ANOVA) and Duncan’s multiple-range test (*p* < 0.05). The correlation analysis and principal component analysis (PCA) were also conducted using Origin 8.0. All identified volatile compounds were engaged in the principal composition analysis. The gallery plot plug-in was used in GC-IMS analysis for fingerprint comparison, i.e., visual and quantitative comparisons of volatile differences between different samples.

## 3. Results and Discussion

### 3.1. GC-MS Analysis of Volatile Profile of Dried Citrus Peel as Effected by Drying Methods

In present study, by using SPME-GC-MS analysis, a total of 56 volatile compounds were tentatively identified and semi-quantified in dried *Citrus reticulata* “Chachi”/“Ponkan” peel prepared from different drying methods, as listed in Table 1 (the approximate concentration of each compound in samples) and Appendix A (the relative percentage content of each compound in samples through the peak area normalization method). For all 56 compounds, identification was based on the chromatographic peak LRI, similarity index higher than 80%, and the available reference standards. In Table 1, the volatiles could be classified as alcohols, aromatic hydrocarbons and ethers, phenols, *N*-containing compounds, aldehydes, monoterpenes, sesquiterpenes, and esters, with the approximate concentration and relative percentage content of each category shown in Figure 1a,b, respectively. It could be found in Figure 1a that, for the total approximate concentration of all detected compounds, there was the order of SD > AD > FD for dried peel of both Chachi and Ponkan. Thus, it could be deduced that sun-drying may partly account for the quality of dried Chachi peel.

#### 3.1.1. Alcohols

As shown in Table 1, the alcohols identified in dried citrus peel were mainly terpenic alcohols, with 2-methyl-3-buten-2-ol only identified in GFD and β-copaen-4α-ol only identified in GAD/GSD. Moreover, linalool and α-terpineol were the main alcohols in both cultivars, with the highest content of linalool and α-terpineol observed in FSD and GSD, respectively. In addition, for both cultivars, SD and AD showed much higher content of linalool and α-terpineol than FD, demonstrating the unfavourability of FD to preserve alcohols in dried citrus peel. Since FD was also unfavorable to keep other categories of volatile compounds, the percentage of alcohols in volatiles of freeze-drying citrus peel was no less than citrus peel made from SD and AD, as shown in Figure 1b. Thus, the combination of content and relative percentage was necessary to investigate the effect of drying methods on the volatile composition.

Since much isomers were detected in volatiles of dried citrus peel, the extracted ion chromatogram (EIC) of different isomers is shown in Figure 2, to more clearly demonstrate the chemical composition of dried citrus peel as influenced by drying methods for the two cultivars. Figure 2a shows the chromatograms of three terpenic alcohol isomers (C_10_H_18_O: linalool, terpinen-4-ol, α-terpineol) through EIC of *m/z* 136. It can be seen from Figure 2a that dried Chachi peel showed much higher content of α-terpineol and terpinene-4-ol (both cyclic terpenic alcohol), especially for SD; and smaller content of linalool (acyclic terpenic alcohol). Thus, the ratio of cyclic to acyclic terpenic alcohol might be a potential indictor to discriminate dried Chachi peel and dried Ponkan peel.

#### 3.1.2. Aromatic Hydrocarbons and Ethers

Two aromatic hydrocarbons (β-cymene, p-cymenene) and one aromatic ether (2-isopropyl-5-methylanisole) were detected in the two cultivars. For both cultivars, it can be found from Table 1 that p-cymenene were only detected in dried Ponkan peel and SD was the best to preserve β-cymene.

#### 3.1.3. Phenols

In Table 1, four phenols (p-thymol, thymol, carvacrol and 2-methoxy-4-vinylphenol) were detected in two cultivars, and FD citrus peel showed the largest peak area and percentage of phenols as compared with dried citrus peel from other two drying methods, suggesting the favorability of FD to preserve phenols in dried citrus peel. Previous reports showed that phenols were bound volatiles with 2-methoxy-4-vinylphenol of the most abundant bound volatile compound in the citrus fruits [31]. In this experiment, phenols were detected in free fractions, and the main phenol in dried Chachi and Ponkan peel were different, with 2-methoxy-4-vinylphenol as the main phenol in FFD and thymol as the main phenol in GFD. Moreover, for the three phenol isomers (p-thymol, thymol, carvacrol), the EIC of *m/z* 150 (Figure 2b) could clearly demonstrate the composition of phenol isomers (C_10_H_14_O: MW 150) as influenced by drying methods for the two cultivars: FD was the best to preserve all detected phenols in dried citrus peel; and dried Chachi peel showed more categories and content of phenol isomers (C_10_H_14_O) than dried Ponkan peel.

#### 3.1.4. *N*-Containing Compounds

There were many reports on the presence of methyl methanthranilate in dried Chachi peel. In this experiment (Table 1), despite of different drying methods, methyl methanthranilate was detected both in dried Chachi and Ponkan peel with the former of much higher content (as can be seen in Figure 1). The other two *N*-containing compounds (methyl anthranilate, 4-acetyl-3,5-dimethyl-2-pyrrolecarboxylic acid, methyl ester) were only detected in dried Chachi peel with low content, suggesting the content and number of *N*-containing compounds could be used to control the quality of dried Chachi peel, in agreement with the previous report [22]. The same effect of drying methods on the percentage of *N*-containing compounds was observed for the two cultivars: FD > AD > SD. However, for the content of *N*-containing compounds, there was no significant effect of drying methods on dried Ponkan peel, while SD and AD were better to preserve *N*-containing compounds than FD for dried Chachi peel.

#### 3.1.5. Aldehydes

The aldehydes detected in dried citrus peel in this experiment include aromatic aldehydes (furfural, benzaldehyde), cyclic aldehyde (perillal), saturated aliphatic aldehydes (octanal, decanal, undecanal, dodecanal and tetradecanal), and unsaturated aliphatic aldehydes (β-sinensal, α-sinensal, 7-pentadecene-7-carbaldehyde). Aliphatic aldehydes, such as octanal, decanal, dodecanal, and sinensal were reported to show significant aroma activity in citrus fruit oil [32,33]. It can be found from Table 1 that dried Ponkan peel showed higher content of both aromatic and aliphatic aldehydes than dried Chachi peel except for α-sinensal and perillal, and dried Chachi peel contains much higher content of α-sinensal (the most aldehyde detected in dried Chachi peel) than dried Ponkan peel; thus, total aldehydes were observed to be higher in dried Chachi peel. For the peel of both cultivars, FD was the best to preserve aromatic aldehydes while AD was the best to preserve aliphatic aldehydes.

#### 3.1.6. Monoterpenes

In agreement with previous reports [16], monoterpenes (C_10_H_16_: MW 136) were often the main compounds in volatiles of dried citrus peel except for the FD peel, as shown in Figure 1a; and among all monoterpenes, D-limonene showed the highest content, followed by γ-terpinene. β-Myrcene and D-limonene were characterized as the potent odorants in the volatile fraction from Pontianak orange peel and wild citrus Mangshanyegan peel oil [34,35]. Under different drying methods, dried Chachi peel always showed the smaller peak area ratio of D-limonene/γ-terpinene than dried Ponkan peel, in accordance with the preliminary review [36]. Moreover, if dried Ponkan peel was made from SD rather than the often-used SD, the ratio of D-limonene/γ-terpinene could also be used to discriminate dried Chachi and Ponkan peel, thus suggesting again that the ratio of D-limonene/γ-terpinene could be used to identify dried Chachi peel. In Table 1 and Figure 1, it can be also found that SD was the best way to preserve monoterpenes, followed by AD, and FD was unfavorable to preserve monoterpenes. Moreover, to clearly demonstrate the monoterpene composition as influenced by drying methods in the two cultivars peel, the EIC (*m/z* 136) of all dried peel are also shown in Figure 2c, which clearly shows that the content and number of monoterpenes was highest in dried Chachi peel, especially for GSD, suggesting that monoterpenes were the characteristic compounds of dried Chachi peel and, again, SD may account for the quality of dried Chachi peel.

#### 3.1.7. Sesquiterpenes

For sesquiterpenes (C_15_H_24_: MW 204), they were also the other important part in volatiles of dried citrus peel, and some sesquiterpenes, such as copaene, β-elemene, caryophyllene and farnesene were common in different citrus germplasms [17,37]. The sesquiterpenes were reported to have efficacy in defending against insects and fungus [38,39]. As shown in Figure 1a and Table 1, FD was also unfavorable to preserve sesquiterpenes; however, AD was the best way to preserve sesquiterpenes, which was unlike monoterpenes. The EIC (*m/z* 204) of all dried peel are also shown in Figure 2d to clearly demonstrate that the sesquiterpene composition is influenced by drying methods in the two cultivars. It can be seen from Figure 2d that the content and number of sesquiterpene isomers was highest in AD peel for both two cultivars. Moreover, each cultivar peel had its characteristic sesquiterpenes, such as γ-elemene, trans-β-farnesene and germacrene B only in dried Ponkan peel while α-humulene and (Z, E)-α- farnesene only in dried Chachi peel; and dried Chachi peel contains much higher content of α-farnesene (the most sesquiterpene detected in dried Chachi peel) than dried Ponkan peel, resulting in a higher level of total sesquiterpenes in dried Chachi peel.

#### 3.1.8. Esters

In Table 1, four unsaturated fatty acid esters (neryl acetate, methyl palmitoleate, methyl linoleate, and methyl linolenate) and one saturated fatty acid ester (methyl palmitate) were detected in dried peel of the two cultivars. Neryl acetate was a major monoterpene ester in most citrus germplasms [17] and showed decreased content in the oleocellosis peel of orange [40]. For both cultivar peels, it can be found from Table 1 that straight-chain fatty acid methyl esters (methyl palmitoleate, methyl palmitate, methyl linoleate, and methyl linolenate) were only detected in FD peel with saturated fatty acid methyl ester (methyl palmitate) of highest peak area and percentage; while branched monoterpene ester (neryl acetate) was undetected in FD peel. Then, for both cultivars, esters accounted for about 20% for FD peel while it had an almost negligible percentage for AD/SD peel (Appendix A and Figure 1b). Thus, there were few reports on straight-chain fatty acid methyl esters in dried citrus peel since citrus peel was always dried by AD or SD.

#### 3.1.9. Correlation and PCA Analysis

For overall analysis of all detected volatiles by GC-MS in dried citrus peel from different drying methods, correlation and PCA analysis was employed in Table 1 with results shown in Figure 3, Figure 4 and Appendix A, respectively. In Figure 3, the Pearson’s correlation coefficient between samples showed that the triplicate samples of the same citrus peel with the same drying method were highly correlated, indicating the good repeatability of the samples within the group. Sample correlation analysis also showed that the differences between FAD and FSD, GAD and GSD were the smallest, followed by the differences between GSD and FAD, GSD and FSD, GFD and FFD, GSD and GFD, and the differences between GFD and FAD, GFD and FSD were the largest. In PCA analysis, the first three principal components covered 88.54% of total variance (principal component-1 = 46.30%, principal component-2 = 30.70%, principal component-3 = 11.54%). Figure 4 shows the first two principal component score/loading plots (covering 77.00% of total variance) while the plots of principal component-1 vs. principal component-3 and principal component-2 vs. principal component-3 are shown in Appendix A. It can be directly observed from Figure 4a and Appendix Aa,c that both drying methods and cultivars showed great influences on the volatile composition of dried citrus peel, and SD peel was more similar with AD peel in GC-MS analysis of volatiles.

In Figure 4b and Appendix A, all three *N*-containing compounds, four sesquiterpenes ((-)-β-caryophyllene, (Z, E)-α-famesene, α-selinene, α-farnesene), and thymol were of positive higher loadings at principal component-1, while five aldehydes (dodecanal, β-sinensal, furfural, decanal, octanal) and four sesquiterpenes (germacrene B, γ-elemene, trans-β-famesene, δ-elemene) were of negative higher loadings at principal component-1, suggesting these compounds represented most important information of principal component-1. Based on the interpretation of principal component-1, it can be seen from Figure 4a and Appendix A that *N*-containing compounds, sesquiterpenes, and aldehydes were important compounds to clearly discriminate dried Chachi and Ponkan peel, with both possessing its individual characteristic sesquiterpenes and aldehydes, and much higher content of *N*-containing compounds in dried Chachi peel.

For principal component-2, as shown in Figure 4b and Appendix A, two monoterpenes (D-limonene and β-myrcene) and two alcohols (linalool and elemol) were of positive higher loadings, while four straight-chain fatty acid methyl esters (methyl palmitoleate, methyl palmitate, methyl linoleate and methyl linolenate) and two phenols (p-thymol, carvacrol) were of negative higher loadings, suggesting these compounds represented most important information of principal component-2. Based on this interpretation, it can be seen from Figure 4a and Appendix A that monoterpenes (including the main compound D-limonene), straight-chain fatty acid methyl esters and phenols were important compounds to clearly discriminate FD peel and AD/SD peel, with FD favorable to preserve phenols and esters while unfavorable to preserve monoterpenes when being compared with AD or SD.

### 3.2. OAVs and Sensory Attributes

The OAV value can be used to measure the contribution of aroma compounds to the overall aroma of the sample. The OAVs of the aroma compounds detected by GC-MS analysis, together with their odor threshold values, are listed in Table 2.

For dried Ponkan peel, in FAD, the OAVs for linalool, 2-methoxy-4-vinylphenol, decanal, perillal, dodecanal, and D-limonene were all beyond 1000, with 2-methoxy-4-vinylphenol of the highest at 12,493, followed by linalool at 6086; in FSD, the OAVs for linalool, 2-methoxy-4-vinylphenol, perillal, and D-limonene were all beyond 1000, with 2-methoxy-4-vinylphenol of the highest at 9028, followed by linalool at 6254; in FFD, the OAVs for linalool, 2-methoxy-4-vinylphenol, and dodecanal were all beyond 1000, with 2-methoxy-4-vinylphenol of the highest at 52,323. It can be seen that 2-methoxy-4-vinylphenol contributed much to the flavor of dried Ponkan peel, especially for the sample prepared from FD.

For dried Chachi peel, in GAD, the OAVs for linalool, 2-methoxy-4-vinylphenol, methyl anthranilate, methyl methanthranilate, perillal, and α-sinensal were all beyond 1000, with methyl methanthranilate of the highest at 9722, followed by 2-methoxy-4-vinylphenol at 4050 and methyl anthranilate at 3913; in GSD, the OAVs for linalool, 2-methoxy-4-vinylphenol, methyl anthranilate, methyl methanthranilate, perillal, α-sinensal, D-limonene, γ-terpinene and terpinolene were all beyond 1000, with methyl methanthranilate of the highest at 9674, followed by methyl anthranilate at 7336; in GFD, the OAVs for 2-methoxy-4-vinylphenol, methyl anthranilate, and methyl methanthranilate were all beyond 1000, with methyl anthranilate of the highest at 7872, followed by methyl methanthranilate at 6699 and 2-methoxy-4-vinylphenol at 4883. It can be seen that, except for 2-methoxy-4-vinylphenol, methyl methanthranilate and methyl anthranilate also contributed much to the flavor of dried Chachi peel, especially for the sample prepared from AD and SD.

Despite of different drying methods, the higher OAVs for 2-methoxy-4-vinylphenol and linalool were observed for dried Ponkan peel while the higher OAVs for methyl anthranilate, methyl methanthranilate, perillal, α-sinensal, γ-terpinene, and terpinolene were observed for dried Chachi peel. The OAVs indicated that these compounds could differentiate the aroma of dried Ponkan and Chachi peel. Furthermore, for both dried Ponkan and Chachi peel, the highest OAVs for monoterpenes were both observed for the SD, indicating monoterpenes may be the characteristic aroma compounds of SD citrus peel. In addition, although D-limonene was the most abundant compound among the volatile components of citrus fruits, its contribution to the aroma was not so much, in accordance with the literature [41].

### 3.3. GC-IMS Analysis of Volatile Profile of Dried Citrus Peel as Effected by Drying Methods

This study also adopted HS-GC-IMS to determine the volatiles in dried citrus peel of two cultivars prepared from different drying methods. The flavor compounds were characterized by comparing their drift time and retention index from the HS-GC-IMS spectrum (Appendix A) with results shown in Table 3. In Table 3, 60 typical flavor compounds were identified from the topographic maps of dried citrus peel via GC-IMS library searches, and some individual compounds generated several signals due to the existence of different polymerization. Moreover, the identified compounds listed in the gallery plot were combined to show the characteristic flavor fingerprints of different dried Chachi and Ponkan peels in Figure 5, where one row represents one sample, and one column represents the intensity of the compound in different samples. A brighter signal of each compound indicates stronger signal intensity, while a darker signal indicates weaker signal intensity.

As shown in Figure 5 and Table 3, 14 alcohols, 16 aldehydes, 4 esters, 11 ketones, 4 monoterpenes, 3 acids, and 8 other compounds were detected in volatiles of different dried citrus peel by GC-IMS analysis. By comparing Table 1 and Table 3, it can be found that most detected compounds were different between GC-MS and GC-IMS analyses, suggesting that GC-MS and GC-IMS analyses are complementary in volatile analysis, in agreement with the literature [42].

#### 3.3.1. Aldehydes and Alcohols

Aldehydes were the most widely detected compounds in dried citrus peel, with 16 aldehydes detected in Figure 5b. Among these aldehydes, most aldehydes contained no more than 10 carbons. The aldehydes were reported from degradation of fatty acid [43]. It was shown that aldehydes were important odorants, imparting the flavor of grass, fruit, almond, and sweet [44]. Hexanal and (Z)-3-hexenal were reported to be important odorants in the headspace above the Pontianak peels [35]. Dried Ponkan peel showed higher concentration of furfural and benzaldehyde than dried Chachi peel, in agreement with GC-MS analysis; while dried Chachi peel showed higher concentration of 3-methylbutanal (malt) and (E)-2-heptenal (soap, fat, almond [44]) than dried Ponkan peel. AD was unfavorable to preserve (E)-2-pentenal (strawberry, fruit, tomato [44]).

The detected alcohols were only smaller than aldehydes with 14 alcohols detected as shown in Figure 5a. Alcohols were also important odorants in citrus fruit: linalool and 1,8-cineole were characterized as the most potent odorant in citrus fruit oil [32,33,35]. Linalool, with the brightest points (implying high concentration) in GC-IMS analysis, was also detected in GC-MS analysis, and higher concentration of linalool was observed in dried Ponkan peel through linalool-polymer point, in agreement with GC-MS analysis. It can be also found from Figure 5a that n-hexanol (resin, flower, green [44]) was characteristic for SD peel; and FD was favorable to preserve 2-methyl-1-butanol (wine, onion [44]).

#### 3.3.2. Monoterpenes

According to the brightness of points, it can be found that monoterpenes in Figure 5c were of the highest content in different dried citrus peels, in accordance with GC-MS data in Table 1. Monoterpenes impart typical aromas of citrus, mint and turpentine, contributing the aromatic notes in orange peel oil [35,45]. However, different from GC-MS analysis, the effect of drying methods on monoterpenes of dried citrus peels was unclear in GC-IMS analysis, probably because of saturated detection due to much higher concentration of monoterpenes in dried citrus peels and great sensitivity of the IMS detector.

#### 3.3.3. Esters, Ketones, Acids, and Others

Esters, usually of fruity or floral smell [44], are the main components contributing to citrus head scent; and ethyl acetate and ethyl butyrate were often considered as quality indexes of citrus aroma [46]. For esters detected in GC-IMS analysis, although different esters were detected as compared with GC-MS analysis, FD was always the best way to preserve esters.

Ketones and acids were undetected in GC-MS analysis, while 11 ketones and 3 acids were detected in GC-IMS analysis (Figure 5d,f). Ketones impart the odor of butter, cream, and ether. It can be found from Figure 5d that AD was unfavorable to preserve most ketones, including 2,3-butanedione, 2-hexanone, acetophenone, cyclohexanone, and 6-methyl-5-hepten-2-one, which were higher in dried Chachi peel than in dried Ponkan peel. While hydroxyacetone were higher in AD peel since hydroxyacetone was formed through pyrolysis of carbohydrate at high temperature during the AD process [47]. Among the three detected acids (Figure 5f), 2-methylbutanoic acid (cheese, sweat [44]) was higher in GSD/GFD than other dried peels. Overall, SD could preserve ketones (2,3-butanedione, 2-hexanone, acetophenone, cyclohexanone and 6-methyl-5-hepten-2-one) and 2-methylbutanoic acid as FD, and dried Chachi peel possessed higher concentration of these ketones and acid than dried Ponkan peel. Thus, ketones (2,3-butanedione, 2-hexanone, acetophenone, cyclohexanone and 6-methyl-5-hepten-2-one) and 2-methylbutanoic acid could also be considered as the characteristic compounds of dried Chachi peel and the contribution of SD to the quality of dried Chachi peel might be partly from the preservation of these ketones and acid.

For other detected compounds in GC-IMS analysis, including *S*-containing compounds, 2-acetylfuran, 2-ethyl-5-methylpyrazine, styrene, and two phenols, it could be found that 2-acetylfuran was a characteristic compound in GFD; while two phenols were characteristic for dried Chachi peel with much higher concentration in GFD/GSD than GAD, suggesting again that FD was favorable to preserve phenols and SD could preserve these two detected phenols in GC-IMS analysis as FD.

#### 3.3.4. Correlation and PCA Analysis

Correlation and PCA analysis were also employed in GC-IMS analysis with results showing that the samples within the group have good repeatability, and the Pearson’s correlation coefficient between SD and FD was higher than that between SD and AD (as shown in Appendix A); and for both cultivars, SD citrus peel was more similar with FD citrus peel (as shown in Appendix A), suggesting SD could preserve volatile flavor compounds (with low molecular weight) as FD. Thus, GC-IMS analysis could serve as a tool to discriminate dried citrus peel in the market from SD and AD, and GC-IMS analysis again demonstrated the contribution of SD to the quality of dried Chachi peel.

## 4. Conclusions

The volatile compounds of dried Chachi and Ponkan peel prepared from three drying methods were analyzed by GC-MS and GC-IMS with a total of 56 and 60 volatile compounds being detected, respectively. The result of GC-MS indicated that for both cultivars, sun-drying was favorable to preserve terpenic alcohols (linalool, α-terpineol and terpinene-4-ol), β-cymene, methyl methanthranilate and monoterpenes, and hot-air-drying was favorable to preserve aliphatic aldehydes and sesquiterpenes. Moreover, the extracted ion chromatogram of isomers in the GC-MS analysis was useful in illustrating the effect of drying methods on the isomeric composition of volatiles in dried citrus peel. GC-IMS data showed that freeze-drying was the best to preserve esters and phenols; hot-air-drying was unfavorable to preserve most ketones; and n-hexanol was characteristic for sun-drying peel.

In addition, the PCA outcomes demonstrated that for dried citrus peel of both cultivars, sun-drying showed the similar volatile profile (of higher MW) with hot-air-drying through GC-MS analysis, and also the similar volatile profile (of smaller MW) with freeze-drying through GC-IMS analysis, suggesting sun-drying was the best drying method to preserve total volatiles in dried Chachi and Ponkan peel, and sun-drying played an important role in the quality of dried Chachi peel.

Furthermore, the odor activity values indicate that 2-methoxy-4-vinylphenol contributed much to the flavor of dried Ponkan peel, while 2-methoxy-4-vinylphenol, methyl methanthranilate, and methyl anthranilate played an important role in the flavor of dried Chachi peel; and the highest odor activity values for monoterpenes were both observed at hot-air-drying for both cultivars, indicating monoterpenes may be the characteristic aroma compounds of hot-air-drying citrus peel.

## Figures and Tables

**Figure 1 foods-11-02662-f001:**
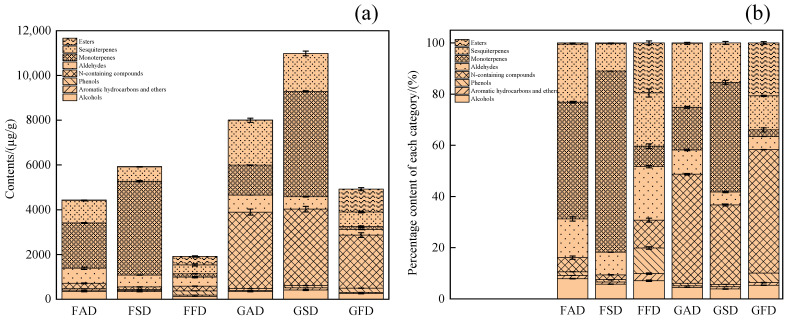
The contents (**a**) and relative percentage (**b**) of each category volatiles in dried Chachi and Ponkan peels prepared from three drying methods.

**Figure 2 foods-11-02662-f002:**
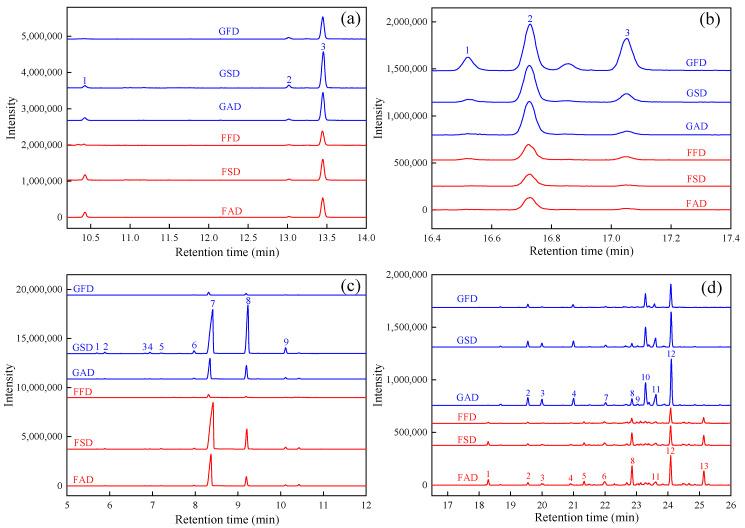
Extracted ion chromatograms (EIC) of *m/z* 136 ((**a**): 1, linalool; 2, terpinene-4-ol; 3, α-terpineol), *m/z* 150 ((**b**): 1, p-thymol; 2, thymol; 3, carvacrol), *m/z* 136 ((**c**): 1, α-thujene; 2, (-)-α-pinene; 3, β-phellandrene; 4, (-)-β-pinene; 5, β-myrcene; 6, α-terpinene; 7, D-limonene; 8, γ-terpinene; 9, terpinolene) and *m/z* 204 ((**d**): 1, δ-elemene; 2, α-copaene; 3, β-elemene; 4, (-)-β-caryophyllene; 5, γ-elemene; 6, trans-β-famesene; 7, α-humulene; 8, D-germacrene; 9, (Z,E)-α-farnesene; 10, α-selinene; 11, α-farnesene; 12, (-)-β-cadinene; 13, germacrene B).

**Figure 3 foods-11-02662-f003:**
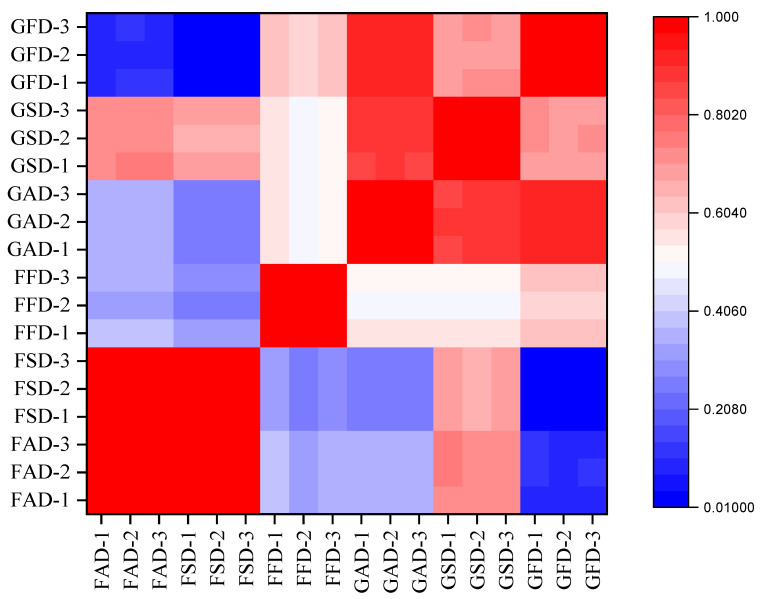
Correlation between dried citrus peel samples of different groups (FAD, FSD, FFD, GAD, GSD, GFD) as characterized by GC-MS analysis.

**Figure 4 foods-11-02662-f004:**
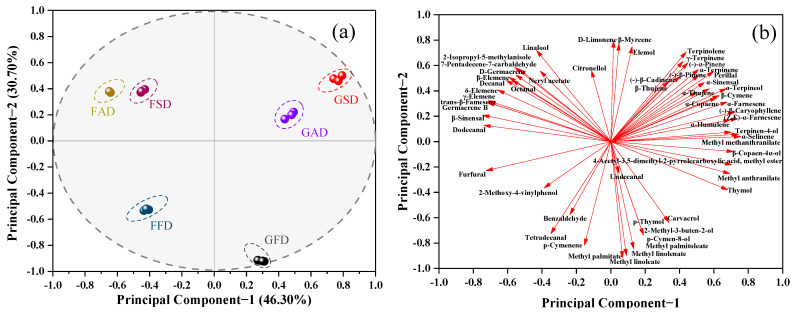
Plots of principal component scores (**a**) and loadings (**b**) in PCA analysis for GC-MS data of dried Chachi and Ponkan peels prepared from three drying methods.

**Figure 5 foods-11-02662-f005:**
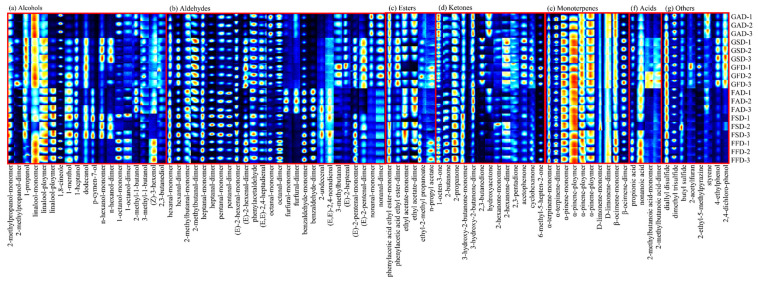
Gallery plot of the selected signal peak areas obtained from volatiles of dried Chachi and Ponkan peels prepared from three drying methods.

**Table 1 foods-11-02662-t001:** The volatile compounds and contents of different dried citrus peel samples by HS-SPME-GC-MS.

Compound	CAS	Identification ^2^	LRI ^1^	Formula	FAD	FSD	FFD	GAD	GSD	GFD
Alcohols										
2-Methyl-3-buten-2-ol	115-18-4	MS + LRI	<800	C_5_H_10_O	nd ^c^	nd	nd	nd	nd	14.84 ± 0.33
Linalool	78-70-6	MS + LRI *	1101	C_10_H_18_O	170.40 ± 4.76 ^a^	175.11 ± 4.40 ^a^	29.35 ± 0.59 ^d^	74.41 ± 2.07 ^b^	67.17 ± 2.08 ^c^	13.27 ± 0.51 ^e^
Terpinen-4-ol	562-74-3	MS + LRI *	1181	C_10_H_18_O	19.18 ± 0.51 ^d^	22.45 ± 0.26 ^d^	22.39 ± 0.48 ^d^	36.18 ± 1.95 ^c^	73.19 ± 5.83 ^a^	45.94 ± 2.23 ^b^
p-Cymen-8-ol	1197-01-9	MS + LRI	1188	C_10_H_14_O	nd	nd	nd	nd	nd	14.24 ± 0.50
α-Terpineol	98-55-5	MS + LRI *	1194	C_10_H_18_O	112.70 ± 3.88 ^d^	116.38 ± 3.58 ^d^	85.02 ± 0.04 ^e^	156.23 ± 4.28 ^b^	215.32 ± 8.42 ^a^	130.94 ± 7.13 ^c^
Citronellol	106-22-9	MS + LRI *	1230	C_10_H_20_O	26.22 ± 1.55 ^a^	15.65 ± 0.94 ^b^	nd	13.36 + 3.11 ^b^	14.09 ± 0.22 ^b^	12.10 ± 2.40 ^b^
Elemol	639-99-6	MS + LRI	1552	C_15_H_26_O	23.46 ± 0.70 ^b^	12.28 ± 0.92 ^d^	nd	29.45 ± 0.69 ^a^	16.78 ± 0.72 ^c^	nd
β-Copaen-4α-ol	124753-76-0	MS + LRI	1588	C_15_H_24_O	nd	nd	nd	43.46 ± 1.43 ^a^	30.73 ± 4.05 ^b^	30.74 ± 0.71 ^b^
				Sum	351.96 ± 11.39 ^b^	341.88 ± 6.38 ^b^	136.77 ± 0.07 ^d^	353.10 ± 12.15 ^b^	417.28 ± 21.32 ^a^	262.07 ± 13.81 ^c^
Aromatic hydrocarbons and ethers										
β-Cymene	535-77-3	MS + LRI *	1027	C_10_H_14_	13.73 ± 0.28 ^d^	16.74 ± 0.41 ^d^	10.58 ± 1.46 ^e^	32.50 ± 0.00 ^b^	97.20 ± 0.71 ^a^	27.63 ± 2.54 ^c^
p-Cymenene	1195-32-0	MS + LRI	1092	C_10_H_12_	nd	nd	41.96 ± 2.28 ^a^	nd	nd	24.85 ± 0.17 ^b^
2-Isopropyl-5-methylanisole	1076-56-8	MS + LRI *	1236	C_11_H_16_O	41.20 ± 2.02 ^a^	31.03 ± 0.12 ^b^	nd	14.64 ± 1.00 ^c^	nd	nd
				Sum	54.93 ± 2.30 ^b^	47.77 ± 0.29 ^c^	52.54 ± 3.74 ^bc^	47.14 ± 1.00 ^c^	97.20 ± 0.71 ^a^	52.48 ± 2.37 ^bc^
Phenols										
p-Thymol	3228-02-2	MS	1287	C_10_H_14_O	nd	nd	nd	nd	nd	21.79 ± 0.65
Thymol	89-83-8	MS + LRI *	1293	C_10_H_14_O	27.81 ± 0.06 ^e^	27.61 ± 2.94 ^e^	35.05 ± 1.79 ^d^	67.64 ± 2.25 ^c^	74.64 ± 3.39 ^b^	91.74 ± 3.04 ^a^
Carvacrol	499-75-2	MS + LRI *	1302	C_10_H_14_O	nd	nd	nd	nd	15.23 ± 2.48 ^b^	53.96 ± 2.48 ^a^
2-Methoxy-4-vinylphenol	7786-61-0	MS + LRI	1316	C_9_H_10_O_2_	37.48 ± 1.58 ^b^	27.08 ± 0.73 ^c^	156.97 ± 0.17 ^a^	12.15 ± 0.14 ^d^	14.41 ± 2.56 ^d^	14.65 ± 1.35 ^d^
				Sum	65.28 ± 1.64 ^e^	54.69 ± 3.67 ^f^	192.02 ± 1.96 ^a^	79.79 ± 2.39 ^d^	104.29 ± 3.32 ^c^	182.14 ± 7.52 ^b^
*N*-containing compounds										
Methyl anthranilate	134-20-3	MS + LRI *	1345	C_8_H_9_NO_2_	nd	nd	nd	11.76 ± 0.37 ^c^	22.01 ± 0.61 ^b^	23.61 ± 0.32 ^a^
Methyl methanthranilate	85-91-6	MS + LRI *	1411	C_9_H_11_NO_2_	242.36 ± 23.30 ^c^	113.66 ± 6.84 ^c^	207.97 ± 6.47 ^c^	3393.14 ± 137.30 ^a^	3376.18 ± 124.59 ^a^	2337.90 ± 102.47 ^b^
4-Acetyl-3,5-dimethyl-2-pyrrolecarboxylic acid, methyl ester	89909-47-7	MS	1644	C_10_H_13_NO_3_	nd	nd	nd	12.58 ± 2.52 ^a^	9.72 ± 0.01 ^a^	11.60 ± 0.76 ^a^
				Sum	242.36 ± 23.30 ^c^	113.66 ± 6.84 ^c^	207.97 ± 6.47 ^c^	3417.47 ± 134.41 ^a^	3407.91 ± 125.20 ^a^	2373.11 ± 102.91 ^b^
Aldehydes										
Furfural	98-01-1	MS + LRI *	832	C_5_H_4_O_2_	69.68 ± 11.16 ^a^	54.02 ± 0.45 ^b^	76.90 ± 6.41 ^a^	28.14 ± 3.56 ^c^	13.30 ± 0.24 ^d^	39.24 ± 0.66 ^c^
Benzaldehyde	100-52-7	MS + LRI *	964	C_7_H_6_O	nd	nd	24.72 ± 0.90 ^a^	nd	nd	8.84 ± 1.89 ^b^
Octanal	124-13-0	MS + LRI *	1004	C_8_H_16_O	12.56 ± 0.20 ^a^	10.97 ± 0.20 ^a^	nd	nd	nd	nd
Decanal	112-31-2	MS + LRI *	1206	C_10_H_20_O	78.74 ± 1.46 ^a^	66.15 ± 0.86 ^b^	36.84 ± 0.17 ^c^	38.24 ± 0.80 ^c^	30.23 ± 0.30 ^d^	30.39 ± 1.84 ^d^
Perillal	2111-75-3	MS + LRI	1277	C_10_H_14_O	30.27 ± 0.03 ^c^	31.44 ± 0.97 ^c^	nd	61.20 ± 2.69 ^b^	68.11 ± 0.43 ^a^	29.54 ± 1.42 ^c^
Undecanal	112-44-7	MS + LRI *	1308	C_11_H_22_O	10.63 ± 0.83 ^a^	nd	nd	10.12 ± 0.37 ^a^	nd	11.48 ± 0.76 ^a^
Dodecanal	112-54-9	MS + LRI *	1407	C_12_H_24_O	80.41 ± 0.91 ^a^	49.26 ± 2.50 ^c^	66.26 ± 0.10 ^b^	nd	nd	nd
Tetradecanal	124-25-4	MS + LRI	1611	C_14_H_28_O	11.51 ± 1.59 ^c^	nd	23.10 ± 0.93 ^a^	nd	nd	15.19 ± 0.43 ^b^
β-Sinensal	60066-88-8	MS + LRI	1700	C_15_H_22_O	136.96 ± 11.34 ^a^	94.58 ± 0.65 ^b^	93.48 ± 7.54 ^b^	nd	nd	nd
α-Sinensal	17909-77-2	MS + LRI	1758	C_15_H_22_O	178.01 ± 17.33 ^c^	140.49 ± 1.45 ^d^	68.38 ± 9.69 ^e^	610.97 ± 6.81 ^a^	428.20 ± 16.82 ^b^	118.09 ± 13.40 ^d^
7-Pentadecene-7-carbaldehyde	-	MS + LRI	1762	C_16_H_30_O	64.42 ± 5.15 ^b^	76.56 ± 2.05 ^a^	11.24 ± 0.97 ^c^	10.92 ± 2.48 ^c^	16.65 ± 0.38 ^c^	nd
				Sum	673.18 ± 49.99 ^b^	523.47 ± 7.01 ^c^	400.93 ± 24.86 ^d^	759.60 ± 1.86 ^a^	556.50 ± 16.22 ^c^	252.77 ± 11.62 ^e^
Monoterpenes										
α-Thujene	2867-05-2	MS + LRI	929	C_10_H_16_	nd	nd	nd	nd	29.81 ± 0.24	nd
(-)-α-Pinene	7785-26-4	MS + LRI *	937	C_10_H_16_	13.07 ± 0.21 ^d^	36.80 ± 0.20 ^b^	nd	18.71 ± 0.21 ^c^	94.63 ± 0.48 ^a^	nd
β-Thujene	28634-89-1	MS + LRI	976	C_10_H_16_	nd	9.46 ± 0.26 ^a^	nd	nd	10.84 ± 0.09 ^a^	nd
(-)-β-Pinene	18172-67-3	MS + LRI *	981	C_10_H_16_	nd	17.60 ± 0.21 ^b^	nd	13.48 ± 0.18 ^c^	71.60 ± 0.02 ^a^	nd
β-Myrcene	123-35-3	MS + LRI *	993	C_10_H_16_	40.43 ± 0.87 ^c^	95.94 ± 0.56 ^a^	nd	21.81 ± 0.08 ^d^	86.64 ± 0.32 ^b^	nd
α-Terpinene	99-86-5	MS + LRI *	1019	C_10_H_16_	nd	16.25 ± 0.60 ^b^	nd	13.24 ± 0.05 ^c^	47.09 ± 1.40 ^a^	nd
D-Limonene	5989-27-5	MS *	1033	C_10_H_16_	1768.96 ± 18.75 ^c^	3599.53 ± 29.64 ^a^	131.65 ± 10.49 ^e^	975.48 ± 8.37 ^d^	3159.90 ± 12.33 ^b^	98.57 ± 29.02 ^f^
γ-Terpinene	99-85-4	MS + LRI *	1061	C_10_H_16_	176.09 ± 2.20 ^d^	382.14 ± 3.12 ^b^	20.90 ± 1.12 ^e^	260.37 ± 0.95 ^c^	1108.98 ± 10.89 ^a^	26.77 ± 5.69 ^e^
Terpinolene	586-62-9	MS + LRI *	1091	C_10_H_16_	23.33 ± 0.41 ^c^	34.07 ± 1.04 ^b^	nd	32.57 ± 0.81 ^b^	93.36 ± 3.94 ^a^	nd
				Sum	2021.88 ± 22.44 ^c^	4191.78 ± 33.17 ^b^	152.54 ± 11.62 ^e^	1335.66 ± 10.12 ^d^	4702.86 ± 28.73 ^a^	125.34 ± 34.71 ^e^
Sesquiterpenes										
δ-Elemene	20307-84-0	MS + LRI	1341	C_15_H_24_	162.62 ± 2.75 ^a^	104.24 ± 2.66 ^b^	36.24 ± 4.84 ^c^	9.05 ± 0.97 ^d^	nd	nd
α-Copaene	3856-25-5	MS + LRI	1380	C_15_H_24_	22.11 ± 0.49 ^c^	12.31 ± 0.23 ^d^	9.81 ± 0.29 ^d^	62.78 ± 3.87 ^a^	47.80 ± 2.29 ^b^	24.78 ± 0.89 ^c^
β-Elemene	515-13-9	MS + LRI	1393	C_15_H_24_	93.99 ± 2.69 ^a^	60.92 ± 2.53 ^b^	29.10 ± 1.75 ^c^	29.92 ± 0.99 ^c^	20.27 ± 0.99 ^d^	13.40 ± 0.76 ^e^
(-)-β-Caryophyllene	87-44-5	MS + LRI	1422	C_15_H_24_	22.28 ± 2.48 ^c^	11.96 ± 1.07 ^c^	9.93 ± 1.02 ^c^	225.01 ± 10.12 ^a^	214.38 ± 13.52 ^a^	94.71 ± 4.94 ^b^
γ-Elemene	29873-99-2	MS + LRI	1433	C_15_H_24_	78.73 ± 1.31 ^a^	52.65 ± 1.03 ^b^	32.88 ± 4.17 ^c^	nd	nd	nd
trans-β-Famesene	18794-84-8	MS + LRI	1454	C_15_H_24_	103.68 ± 1.54 ^a^	64.67 ± 2.22 ^b^	44.62 ± 3.88 ^c^	nd	nd	nd
α-Humulene	6753-98-6	MS + LRI	1456	C_15_H_24_	nd	nd	nd	39.47 ± 2.61 ^a^	28.45 ± 1.78 ^b^	13.90 ± 0.96 ^c^
D-Germacrene	23986-74-5	MS + LRI	1482	C_15_H_24_	143.79 ± 4.65 ^a^	89.04 ± 1.03 ^b^	39.97 ± 5.03 ^d^	49.94 ± 3.12 ^c^	27.32 ± 1.76 ^e^	nd
(Z,E)-α-Farnesene	26560-14-5	MS + LRI	1492	C_15_H_24_	nd	nd	nd	36.96 ± 1.38 ^a^	35.68 ± 1.87 ^a^	16.06 ± 0.63 ^b^
α-Selinene	473-13-2	MS + LRI	1500	C_15_H_24_	nd	nd	16.94 ± 1.60 ^d^	140.77 ± 7.32 ^a^	124.67 ± 6.18 ^b^	74.27 ± 3.95 ^c^
α-Farnesene	502-61-4	MS + LRI *	1510	C_15_H_24_	143.47 ± 0.26 ^d^	88.93 ± 0.20 ^d^	60.82 ± 6.99 ^d^	1291.76 ± 62.20 ^a^	1101.39 ± 67.64 ^b^	353.83 ± 23.95 ^c^
(-)-β-Cadinene	523-47-7	MS + LRI	1527	C_15_H_24_	89.05 ± 3.30 ^b^	59.20 ± 0.67 ^cd^	52.09 ± 3.13 ^d^	119.63 ± 4.85 ^a^	94.10 ± 5.94 ^b^	62.66 ± 3.51 ^c^
Germacrene B	15423-57-1	MS + LRI	1562	C_15_H_24_	147.90 ± 4.10 ^a^	98.68 ± 0.96 ^b^	66.13 ± 14.52 ^c^	nd	nd	nd
				Sum	1007.63 ± 18.60 ^c^	642.60 ± 12.19 ^d^	398.52 ± 46.64 ^e^	2005.29 ± 95.50 ^a^	1694.05 ± 101.97 ^b^	653.59 ± 39.59 ^d^
Esters										
Neryl acetate	141-12-8	MS + LRI *	1365	C_12_H_20_O_2_	20.22 ± 0.00 ^a^	11.60 ± 0.21 ^b^	nd	11.48 ± 0.39 ^b^	nd	nd
Methyl palmitoleate	1120-25-8	MS + LRI	1902	C_17_H_32_O_2_	nd	nd	nd	nd	nd	22.83 ± 1.94
Methyl palmitate	112-39-0	MS + LRI *	1925	C_17_H_34_O_2_	nd	nd	232.79 ± 12.51 ^b^	nd	nd	527.87 ± 37.65 ^a^
Methyl linoleate	112-63-0	MS + LRI *	2094	C_19_H_34_O_2_	nd	nd	100.03 ± 12.25 ^b^	nd	nd	275.04 ± 13.19 ^a^
Methyl linolenate	301-00-8	MS + LRI	2101	C_19_H_32_O_2_	nd	nd	42.04 ± 5.94 ^b^	nd	nd	192.85 ± 15.55 ^a^
				Sum	20.22 ± 0.00 ^c^	11.60 ± 0.21 ^c^	374.86 ± 30.70 ^b^	11.48 ± 0.39 ^c^	nd	1018.60 ± 68.32 ^a^
				Total	4437.44 ± 83.05 ^e^	5927.43 ± 48.32 ^c^	1916.16 ± 78.47 ^f^	8009.52 ± 254.11 ^b^	10980.09 ± 265.03 ^a^	4920.10 ± 206.68 ^d^

^1^ Linear retention indices were calculated according to Majlát et al. [28] ^2^ Identification was performed as follows: MS, mass spectrum was consistent with that in Nist mass spectrum database; LRI, retention index was consistent with that reported in the literature in the Nist database; *, mass spectrum and retention index were consistent with that of authentic compound. The term “nd” means the compound was not detected in a sample. Different lower-case letters in the same row indicate the significant differences (*p* < 0.05). FAD and GAD stand for hot-air-dried peel from Citrus reticulata “Ponkan” (Fujian) and Citrus reticulata “Chachi” (Guangdong); FSD and GSD stand for sun-dried peel from Citrus reticulata “Ponkan” (Fujian) and Citrus reticulata “Chachi” (Guangdong); and FFD and GFD stand for freeze-dried peel from Citrus reticulata “Ponkan” (Fujian) and Citrus reticulata “Chachi” (Guangdong). Same in other tables and figures.

**Table 2 foods-11-02662-t002:** Odor activity values (OAVs) of odorants in dried Citrus peel prepared from different drying methods.

Compound	Threshold in Water from the Literature (mg/kg)	OAVs					
		FAD	FSD	FFD	GAD	GSD	GFD
**Alcohols**							
Linalool	0.028	6086	6254	1048	2658	2399	474
Terpinen-4-ol	6.4	3	4	4	6	11	7
α-Terpineol	10	11	12	9	16	22	13
Citronellol	0.062	423	252	__	216	227	195
Elemol	0.1	235	123	__	295	168	__
**Aromatic hydrocarbons and ethers**							
β-Cymene	0.8	17	21	13	41	121	35
p-Cymenene	0.665	__	__	63	__	__	37
**Phenols**							
Thymol	0.79	35	35	44	86	94	116
Carvacrol	2.29	__	__	__	__	7	24
2-Methoxy-4-vinylphenol	0.003	12,493	9028	52,323	4050	4805	4883
***N*-containing compounds**							
Methyl anthranilate	0.003	__	__	__	3919	7336	7872
Methyl methanthranilate	0.349	694	326	596	9722	9674	6699
**Aldehydes**							
Furfural	3	23	18	26	9	4	13
Benzaldehyde	1	__	__	25	__	__	9
Octanal	0.08	157	137	__	__	__	__
Decanal	0.07	1125	945	526	546	432	434
Perillal	0.03	1009	1048	__	2040	2270	985
Undecanal	0.04	266	__	__	253	__	287
Dodecanal	0.055	1462	896	1205	__	__	__
Tetradecanal	0.067	172	__	345	__	__	227
α-Sinensal	0.22	809	639	311	2777	1946	537
**Monoterpenes**							
(-)-α-Pinene	0.1	131	368	__	187	946	__
(-)-β-Pinene	4.16	__	4	__	3	17	__
β-Myrcene	0.099	408	969	__	220	875	__
α-Terpinene	0.08	__	203	__	166	589	__
D-limonene	1.2	1474	3000	110	813	2633	82
γ-Terpinene	0.6	293	637	35	434	1848	45
Terpinolene	0.041	569	831	__	794	2277	__
**Sesquiterpenes**							
(-)-β-caryophyllene	1.54	14	8	6	146	139	62
α-humulene	0.39	__	__	__	101	73	36

The term “__” means the compound was not detected in sample.

**Table 3 foods-11-02662-t003:** Volatile compounds identified in different dried citrus peel samples by GC-IMS.

Compound	CAS	Formula	MW	RI	RT (s)	DT (ms)	Comment
Alcohols							
2-Methylpropanol	78-83-1	C_4_H_10_O	74.1	655.3	178.498	1.1728	Monomer
2-Methylpropanol	78-83-1	C_4_H_10_O	74.1	638.2	170.541	1.3634	Dimer
1-Propanol	71-23-8	C_3_H_8_O	60.1	562.9	141.808	1.1113	
Linalool	78-70-6	C_10_H_18_O	154.3	1098.9	806.393	1.2181	Monomer
Linalool	78-70-6	C_10_H_18_O	154.3	1099.4	807.109	1.6902	Polymer
Linalool	78-70-6	C_10_H_18_O	154.3	1108.2	821.018	2.2416	Polymer
1-Menthol	2216-51-5	C_10_H_20_O	156.3	1176.5	928.37	1.8827	
1,8-Cineole	470-82-6	C_10_H_18_O	154.3	1024	683.053	1.2952	
1-Heptanol	111-70-6	C_7_H_16_O	116.2	967.5	570.796	1.3964	
Dodecanol	112-53-8	C_12_H_26_O	186.3	1495.5	1430.277	1.6136	
p-Cymen-7-ol	536-60-7	C_10_H_14_O	150.2	1314.5	1145.531	1.3218	
n-Hexanol	111-27-3	C_6_H_14_O	102.2	875.6	394.101	1.3286	Monomer
n-Hexanol	111-27-3	C_6_H_14_O	102.2	874.4	392.236	1.6395	Dimer
1-Octanol	111-87-5	C_8_H_18_O	130.2	1077.4	772.375	1.474	Monomer
1-Octanol	111-87-5	C_8_H_18_O	130.2	1075.3	769.058	1.8822	Dimer
2-Methyl-1-butanol	137-32-6	C_5_H_12_O	88.1	712.1	215.353	1.2361	
3-Methyl-1-butanol	123-51-3	C_5_H_12_O	88.1	732.7	233.387	1.2497	
(Z)-3-Hexenol	928-96-1	C_6_H_12_O	100.2	868.3	383.342	1.2298	
2,3-Butanediol	513-85-9	C_4_H_10_O_2_	90.1	791.1	291.806	1.3693	
Aldehydes							
Hexanal	66-25-1	C_6_H_12_O	100.2	793.9	294.682	1.2569	Monomer
Hexanal	66-25-1	C_6_H_12_O	100.2	794.5	295.284	1.5665	Dimer
2-Methylbutanal	96-17-3	C_5_H_10_O	86.1	671.2	186.926	1.1585	Monomer
2-Methylbutanal	96-17-3	C_5_H_10_O	86.1	664.6	183.314	1.4033	Dimer
Heptanal	111-71-7	C_7_H_14_O	114.2	901.1	435.987	1.3322	Monomer
Heptanal	111-71-7	C_7_H_14_O	114.2	900.4	434.707	1.6959	Dimer
Pentanal	110-62-3	C_5_H_10_O	86.1	699.5	205.405	1.1864	Monomer
Pentanal	110-62-3	C_5_H_10_O	86.1	697.4	203.832	1.4221	Dimer
(E)-2-Hexenal	6728-26-3	C_6_H_10_O	98.1	845.6	352.95	1.1803	Monomer
(E)-2-Hexenal	6728-26-3	C_6_H_10_O	98.1	852.6	361.851	1.5176	Dimer
Phenylacetaldehyde	122-78-1	C_8_H_8_O	120.2	1048.9	726.125	1.2491	
(E,E)-2,4-Heptadienal	4313-03-5	C_7_H_10_O	110.2	1018.1	672.456	1.1866	
Octanal	124-13-0	C_8_H_16_O	128.2	1012.2	661.438	1.4035	Monomer
Octanal	124-13-0	C_8_H_16_O	128.2	1007.9	653.298	1.8273	Dimer
Furfural	98-01-1	C_5_H_4_O_2_	96.1	831.6	335.997	1.0834	Monomer
Furfural	98-01-1	C_5_H_4_O_2_	96.1	830.7	334.948	1.3324	Dimer
Benzaldehyde	100-52-7	C_7_H_6_O	106.1	956.6	547.609	1.1494	Monomer
Benzaldehyde	100-52-7	C_7_H_6_O	106.1	956.6	547.609	1.465	Dimer
2-Undecenal	2463-77-6	C_11_H_20_O	168.3	1380.8	1249.854	1.4952	
(E,E)-2,4-Nonadienal	5910-87-2	C_9_H_14_O	138.2	1240.1	1028.477	1.3537	
3-Methylbutanal	590-86-3	C_5_H_10_O	86.1	655.1	178.408	1.1959	
(E)-2-Heptenal	18829-55-5	C_7_H_12_O	112.2	954.8	543.803	1.2541	
(E)-2-Pentenal	1576-87-0	C_5_H_8_O	84.1	754.1	253.909	1.1022	Monomer
(E)-2-pentenal	1576-87-0	C_5_H_8_O	84.1	754.2	253.99	1.3603	Dimer
Nonanal	124-19-6	C_9_H_18_O	142.2	1115.4	832.379	1.4722	Monomer
Nonanal	124-19-6	C_9_H_18_O	142.2	1113.9	829.928	1.9474	Dimer
Esters							
Phenylacetic acid ethyl ester	101-97-3	C_10_H_12_O_2_	164.2	1225.5	1005.553	1.2959	Monomer
Phenylacetic acid ethyl ester	101-97-3	C_10_H_12_O_2_	164.2	1224.3	1003.65	1.7843	Dimer
Ethyl acetate	141-78-6	C_4_H_8_O_2_	88.1	601.9	156.127	1.0953	Monomer
Ethyl acetate	141-78-6	C_4_H_8_O_2_	88.1	605.4	157.428	1.3376	Dimer
Ethyl-2-methyl propanoate	97-62-1	C_6_H_12_O_2_	116.2	727.2	228.365	1.1956	
n-Propyl acetate	109-60-4	C_5_H_10_O_2_	102.1	733.2	233.845	1.1685	
Ketones							
1-Octen-3-one	4312-99-6	C_8_H_14_O	126.2	995.3	628.789	1.2859	
2-Butanone	78-93-3	C_4_H_8_O	72.1	595.1	153.636	1.0617	
2-Propanone	67-64-1	C_3_H_6_O	58.1	509.2	122.061	1.1191	
3-Hydroxy-2-butanone	513-86-0	C_4_H_8_O_2_	88.1	722.9	224.501	1.0599	Monomer
3-Hydroxy-2-butanone	513-86-0	C_4_H_8_O_2_	88.1	717.7	220.027	1.328	Dimer
2,3-Butanedione	431-03-8	C_4_H_6_O_2_	86.1	633	168.329	1.1685	
Hydroxyacetone	116-09-6	C_3_H_6_O_2_	74.1	638.3	170.599	1.0427	
2-Hexanone	591-78-6	C_6_H_12_O	100.2	786.1	286.633	1.1901	Monomer
2-Hexanone	591-78-6	C_6_H_12_O	100.2	787.2	287.739	1.5056	Dimer
2,3-Pentadione	600-14-6	C_5_H_8_O_2_	100.1	660.2	180.929	1.2261	
Acetophenone	98-86-2	C_8_H_8_O	120.2	1067.6	756.873	1.191	
Cyclohexanone	108-94-1	C_6_H_10_O	98.1	897.6	429.786	1.1484	
6-Methyl-5-hepten-2-one	110-93-0	C_8_H_14_O	126.2	990.1	618.121	1.173	
Monoterpenes							
α-Terpinene	99-86-5	C_10_H_16_	136.2	1017	670.467	1.2204	Monomer
α-Terpinene	99-86-5	C_10_H_16_	136.2	1017.4	671.183	1.7217	Dimer
α-Pinene	80-56-8	C_10_H_16_	136.2	942	516.604	1.217	Monomer
α-Pinene	80-56-8	C_10_H_16_	136.2	940	512.312	1.2954	Polymer
α-Pinene	80-56-8	C_10_H_16_	136.2	938.3	508.735	1.6688	Polymer
α-Pinene	80-56-8	C_10_H_16_	136.2	936.5	505.158	1.7341	Polymer
D-Limonene	5989-27-5	C_10_H_16_	136.2	1040.8	712.473	1.2165	Monomer
D-Limonene	5989-27-5	C_10_H_16_	136.2	1038.9	709.238	1.2987	Polymer
β-Ocimene	13877-91-3	C_10_H_16_	136.2	1057.2	739.969	1.2143	Monomer
β-Ocimene	13877-91-3	C_10_H_16_	136.2	1057.7	740.777	1.6943	Dimer
Acids							
Propionic acid	79-09-4	C_3_H_6_O_2_	74.1	704.6	209.337	1.1119	
Nonanoic acid	112-05-0	C_9_H_18_O_2_	158.2	1279.5	1090.406	1.5442	
2-Methylbutanoic acid	116-53-0	C_5_H_10_O_2_	102.1	890.7	417.914	1.2097	Monomer
2-Methylbutanoic acid	116-53-0	C_5_H_10_O_2_	102.1	890.7	417.914	1.4764	Dimer
Others							
Diallyl disulfide	2179-57-9	C_6_H_10_S_2_	146.3	1090.7	793.377	1.1873	
Dimethyl trisulfide	3658-80-8	C_2_H_6_S_3_	126.3	978.3	593.635	1.2952	
Butyl sulfide	544-40-1	C_8_H_18_S	146.3	1085.6	785.453	1.3011	
2-Acetylfuran	1192-62-7	C_6_H_6_O_2_	110.1	909.8	452.045	1.1171	
2-Ethyl-5-methylpyrazine	13360-64-0	C_7_H_10_N_2_	122.2	1002.1	642.196	1.1768	
Styrene	100-42-5	C_8_H_8_	104.2	904.6	442.242	1.0594	
4-Ethylphenol	123-07-9	C_8_H_10_O	122.2	1198.2	962.542	1.1888	
2,4-Dichloro-phenol	120-83-2	C_6_H_4_Cl_2_O	163	1140.7	872.045	1.1901	

MW represents molecular mass, RI represents the relative retention index calculated using n-ketones C4–C9 as external standard on FS-SE-54-CB column, Rt represents the retention time in the capillary GC column, Dt represents the drift time in the drift tube.

## Data Availability

Data is contained within the article.

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
