# Peer review of "Effect of Drying Methods on Volatile Compounds of Citrus reticulata Ponkan and Chachi Peels as Characterized by GC-MS and GC-IMS"

_foods, 2022, doi:10.3390/foods11172662_

Round 1
Reviewer 1 Report
The submitted article decoded the volatile differences of citrus peel as affected by two cultivars and three drying methods. The article is comprehensive and well written, and I suggest the following minor corrections:
Pages 1-2, please provide some literature about potential applications of dried citrus peel.
Page 2, please consider moving 2.2. as 2.1., write first about the plant material, and afterward about reagents and chemicals,
Page 3, line 112, specify the device used for finely grounding the samples
Page 4, specify the type, producer, city, and country for the ionization chamber
Highlight in how many repetitions analyses were performed.
Table 1, please write letters indicating the statistically significant difference as superscript
Figure 1, add standard deviation in figures 1 a and 1 b
Page 4, please consider adding correlation analysis before PCA
Page 8, Figure 4, please use larger font the text is barely visible
Page 10, avoid using abbreviations in the conclusion section
Reviewer 2 Report
Introduction
line 31-35- this text should be supported by statistics of some organizations
Line 36-39- senstence needs citations
The authors should mention the danger of using citrus peel. The peel very often contains all sorts of residues of pesticides or illegal substances. They are used during the growth of the fruit, ripening and transportation, which is a common cause of citrus fruit transport stoppage.
The authors should add a short paragraph regarding the use of PCA to classify food, showing changes in food/raw material under the influence of technological processes, or the identification of significant organic substances in raw materials or food products. Based on the entirety of this article, the authors should add such a paragraph to expose the importance and usefulness of the statistical tool they use in their research. Here I provide some literature in this subject: https://doi.org/10.1111/ijfs.14697 ; https://doi.org/10.1007/s00217-022-04037-4; https://doi.org/10.1016/j.jpba.2019.01.010 ; https://doi.org/10.3390/antiox10101637.
Materials and methods
2.4.1. HS-SPME-GC–MS Analysis - Is this the authors' own methodology or did they use an already published methodology? If a previously published methodology was used then it should be cited as reference. Even if the authors made modifications to it.
On what basis did the authors know that 2 g of sample and the conditioning and extraction of volatile compounds under the reported conditions and specific SPME fiber would be sufficient to this experiment?
line 129- the content of individual volatile compounds on the basis of an internal standard is a semiquantitative analysis. So please change word “quantified” on “semiquantified”.
2.5. Statistical analysis - Please write here whether all identified volatile compounds were engaged in the principal composition analysis. On what basis were volatile compounds classified to participate in this PCA analysis?
Results and discussion
line 166-169- please delete this long sentence, you have already mentioned it in the introduction and methodology so why repeat this information again?
Tables – authors use abbreviations: FAD, FSD, FFD, GAD, GSD, GFD. I only recognize pieces of abbreviations that specify the type of citrus peel drying, but I don't know which fruit the first letter (F or G) in the abbreviation stands for. All abbreviations should be explained in the footer of the table. The same situation is on Figures.
3.1.9. PCA analysis - 77% of the total variance explained by the 1st and 2nd principal components is not enough to write that it is enough to interpret the acquired data. Please add charts with 1 and 3, 2 and 3 of the principal components in the supplement material and describe them in the text of the article.
Table 3 - The authors gave a description of the odor in the table next to each volatile compound, but did not do olfactometric analysis. Therefore, the column in the table describing how the compound smells should be removed.
Conclusions
Conclusions arise from the authors' research. However, they should be divided into specific paragraphs and shortened to convey these key conclusions rather than a summary of the results.
Reviewer 3 Report
The article is interesting and brings new information to the current state of knowledge.
Overall, the article is well prepared. The title and abstract are concise and inform well about the subject and results of the experiment.
The keywords, while appropriate, are nevertheless a repetition of the title. In addition, these should be phrases, the most important keys that the reader can use to find the article in search engines. The authors are requested to correct these inaccuracies.
The introduction is synthetic and introduces the reader to the research issues.
The methodology is properly selected. The authors described the analytical process in great detail.
The results fully reflect the results of the experiment. Authors interpret and discuss their findings with those of other work.
The summary fully reflects the research problem and presents the most important achievements of the presented research results.
In my opinion, the article can be considered in its current form.
Author Response
Dear reviewer,
Thanks you for your suggestion. The keywords were revised as “Citrus peel; volatile compounds; sun drying (SD); hot-air drying (AD); freeze drying (FD); GC-MS; GC-IMS” in the revised manuscript.